# Advanced Glycation End-Products Acting as Immunomodulators for Chronic Inflammation, Inflammaging and Carcinogenesis in Patients with Diabetes and Immune-Related Diseases

**DOI:** 10.3390/biomedicines12081699

**Published:** 2024-07-31

**Authors:** Chieh-Yu Shen, Cheng-Hsun Lu, Chiao-Feng Cheng, Ko-Jen Li, Yu-Min Kuo, Cheng-Han Wu, Chin-Hsiu Liu, Song-Chou Hsieh, Chang-Youh Tsai, Chia-Li Yu

**Affiliations:** 1Department of Internal Medicine, National Taiwan University Hospital, National Taiwan University College of Medicine, # 7 Chung-Shan South Road, Taipei 10002, Taiwan; chiehyushen@ntu.edu.tw (C.-Y.S.); b89401085@ntu.edu.tw (C.-H.L.); chiaofengcheng@gmail.com (C.-F.C.); dtmed170@gmail.com (K.-J.L.);; 2Institute of Clinical Medicine, National Taiwan University College of Medicine, # 7 Chung-Shan South Road, Taipei 10002, Taiwan; 3Department of Internal Medicine, National Taiwan University Hospital-Hsinchu Branch, # 2, Section 1, Shengyi Road, Hsinchu County 302058, Taiwan; chenghanwu@ntu.edu.tw; 4Department of Internal Medicine, National Taiwan University Hospital-Yunlin Branch, # 579, Section 2, Yunlin Road, Yunlin County 640203, Taiwan; 2001windchild@gmail.com; 5Department of Internal Medicine, Fu-Jen Catholic University Hospital, College of Medicine, Fu-Jen Catholic University, # 69 Guizi Road, New Taipei City 24352, Taiwan

**Keywords:** non-enzymatic Maillard reaction, advanced glycation end-products (AGEs), toxic AGE, Diabetes mellitus, AGE binding receptor, lifestyle-related disease, immune-related disease, inflamm-aging, carcinogenesis

## Abstract

Increased production of advanced glycation end products (AGEs) among reducing sugars (glucose, fructose, galactose, or ribose) and amino acids/proteins via non-enzymatic Maillard reaction can be found in lifestyle-related disease (LSRD), metabolic syndrome (MetS), and obesity and immune-related diseases. Increased serum levels of AGEs may induce aging, diabetic complications, cardiovascular diseases (CVD), neurodegenerative diseases (NDD), cancer, and inflamm-aging (inflammation with immunosenescence). The Maillard reaction can also occur among reducing sugars and lipoproteins or DNAs to alter their structure and induce immunogenicity/genotoxicity for carcinogenesis. AGEs, as danger-associated molecular pattern molecules (DAMPs), operate via binding to receptor for AGE (RAGE) or other scavenger receptors on cell surface to activate PI3K-Akt-, P38-MAPK-, ERK1/2-JNK-, and MyD88-induced NF-κB signaling pathways to mediate various pathological effects. Recently, the concept of “inflamm-aging” became more defined, and we have unveiled some interesting findings in relation to it. The purpose of the present review is to dissect the potential molecular basis of inflamm-aging in patients with diabetes and immune-mediated diseases caused by different AGEs.

## 1. Introduction

It has been reported that habitual or chronic consumption of sugar-sweetened beverages (such as high fructose corn syrup), foods, or sucrose are implicated in the development of obesity and metabolic syndrome (MetS). These lifestyle-related diseases (LSRD) can eventually develop into Diabetes mellitus (DM), with its attendant complications [1,2], including cardiovascular diseases (CVD) [3,4], Alzheimers disease (AD) [5,6], non-alcoholic steatohepatitis (NASH) [7,8], and even cancers [9,10]. Although the underlying mechanisms have not been completely explored, the formation of advanced glycation end-products (AGEs) via the Maillard reaction (MR) [11,12] between reducing sugars (glucose, fructose, galactose, ribose) or glyceraldehyde (GA) and amino acids/proteins have been confirmed. Among these, the GA-derived AGEs (GA-AGEs) directly exerted toxicity on cells and are referred to as toxic AGEs (TAGEs) [13]. These intracellular TAGEs can trigger cell damage via the TAGE-RAGE-ROS system, the so called “TAGE theory”. These pathological processes have been observed in LSRD [14,15]. In addition, many cytotoxic compounds, including triosidines, GA-derived pyridinium compounds, GA-derived pyrrolopyridinium lysine dimers, methylglyoxal-derived hydro- imidazolone 1, and argpyrimidine, have already been identified, by biochemical analysis, in the TAGE structures derived from glyceraldehyde [16]. In addition to TAGEs, which are highly toxic and have a relatively high molecular weight, another group of AGEs, which are non-toxic with a relatively low molecular weight, belong to a heterogeneous complex compound formed either exogenously or endogenously. These compounds are synthesized by way of non-enzymatic condensation between carbonyl groups of reducing sugars and free amino groups of proteins, nucleic acids, or lipids via MR. These processes finally yield stable and irreversible end-products, known as advanced glycation end-products [17].

These chemically diverse AGEs have been identified in milk and diary products, as a result of cigarette smoking, and in thermally processed foods. Federio et al. [18] measured AGE-linked skin fluorescence in 101 infants and found that formula-fed infants had increased AGE levels when compared to breast-fed infants. Breast-fed infants from mothers who smoked cigarettes during pregnancy and lactation had higher AGE levels than non-smoking mothers. This evidence reinforces that smoking must be stopped during these periods. For comparing the thermally processed diets to the risk of diabetic complications, Wu et al. [19] conducted high dietary AGE feeding with 8-week-old mice and found that the high AGE exposure inhibited carbohydrate metabolism and promoted lipid anabolism, alteration of gut microbiota, and increased plasma levels of glyceraldehyde and pyruvate. These results indicate that exposure to dietary AGEs modulates carbohydrate and lipid metabolism for diabetes development. 

Takata et al. [20] discovered that blood TAGE levels were elevated in patients with CVD. The same group further examined the cytotoxicity of the intracellular TAGE by a slot analysis. They noted that TAGE could induce human cardiac fibroblasts cell death. Epidemiological studies revealed that diabetic patients were at a high risk for developing Alzheimers disease (AZ). Ooi et al. [21] demonstrated that TAGE could induce TAGE-β tubulin aggregation and tau protein phosphorylation in human neuroblastoma SH-SY5Y cells, leading to dysfunctional neurite growth. Pan et al. [22], in a cohort study, found cancer metastasis incidence was significantly correlated with serum AGEs concentration in patients with breast cancer. During the follow-up, the metastatic interval was significantly shorter in diabetic than non-diabetic patients. In an in vitro study, the group unveiled that AGE-BSA significantly activated cancer cells’ RAGE-TLR4 signaling, thereby promoting breast cancer cell metastasis.

Cell/tissue injury induced by reactive oxygen intermediates is implicated in the development of diabetic vascular damage. Yan et al. [23] demonstrated that AGE-RAGE interaction could induce oxidant stresses in endothelial cells to generate thiobarbituric acid reactive substances (TBARS), resulting in NF-κB activation and vascular lesions. Later, Schmidt et al. [24] further confirmed that infusion of AGE-BSA into mice resulted in increasing generation of TBARS in the gingiva as well as in the lung, kidney, and brain. Pre-treatment of the animals with N-acetylcysteins (NAC) prevented TBARS in the tissues. These data may suggest that enhanced oxidant stress is a potential mechanism for tissue injury in diabetes. Recently, Xiao et al. [25] tried to explore the molecular mechanism of AGE-induced diabetic vasculopathy. The group verified that a multi-functional actin-binding protein, profilin-1, was involved in AGE-induced atherosclerosis via JAK2 and STAT3 signaling for inflammation and vascular remodeling. The same group then injected AGE-BSA into Sprague–Dawley rat. They confirmed that the exogenous AGEs can mimic diabetic vasculopathy in vivo by increased profilin-1 expression in the aorta, heart, and kidneys, and also induced high levels of ICAM-1, IL-8, TNF-α, ROS, and apoptosis after AGE-BSA injection [26]. The source, formation, and classification of AGEs in mediating AGE-related diseases are illustrated in Figure 1.

## 2. High Blood Sugar as a Modifiable Environmental Factor Inducing Persistent Epigenetic Changes in Patients with Glycemic Trait for Insulin Resistance

Many authors had found that amino acids [27], carbohydrates [28], and organic acids [29] were associated with type 2 diabetes (T2DM) pathogenesis, in that these nutrients, via gluconeogenesis, can increase glucose levels. As such, insulin secretion is promoted by the pancreas and exerts a negative impact on pancreatic islet function in long-term gluconeogenesis [30]. Recently, Sun et al. [31] used Mendelian randomization (MR) to examine the cause-effect relationship via a metabolomic study. These authors explored that arachidonic acid (AA) and 2-hydroxybutyric acid (2-HBA) had a positive causal effect on glycemic trait in addition to 1,5-anhydrosorbital (1,5-AG) and hydroxybutyric acid (2-HBA) as novel biomarkers for T2DM [28,32].

In addition to the above-mentioned interrelationship between metabolites and T2DM, the transient hyperglycemia can cause persistent epigenetic changes, thereby altering gene expression even after long-term subsequent normoglycemia [33,34]. Recently, AGEs have been regarded as modifiable environmental factors for promoting oxidative damage to proteins, lipids, and nucleic acids through oxidative stress to elicit insulin resistance as shown in Figure 2. These AGEs can stimulate innate immune cells to increase proinflammatory cytokines production [35,36]. 

## 3. Mechanisms of AGEs as a Major Source of Oxidants Implicating in Diabetic Nephropathy, Tissue Inflammation and Aging

### 3.1. The Link of Circulating AGEs to Oxidative Stress, Inflammatory Response, Aging, and Age-Related Diseases

It is conceivable that oxidative stress (OS) increases with aging [37,38,39] and plays a pathophysiological role in chronic diseases prevalence in elder people, including CVD, renal disease, and DM [40,41,42]. Uribarri et al. [43] further confirmed that the excessive consumption of AGEs via the diet enhanced OS and inflammatory responses in healthy adults. AGEs can also exert profound prognostic effects on chronic kidney disease via persistent inflammation [44,45].

### 3.2. The Interaction of AGEs with Its Receptor RAGE in Inducing Inflammation via ROS Formation as Reflected by Increasing p66shc Protein Expression

The receptor for advanced glycation end-products (RAGE) is a trans-membrane multi-ligand receptor belonging to the immunoglobulin superfamily. RAGE can not only bind with AGEs but also with several ligands, including HMGB1 (high-mobility group box-1), members of the S100 protein family, glycosaminoglycans, and amyloid β peptides. Ligation of RAGE by a pathological factor would activate various cell signals for cell transcription, migration, and acute and chronic inflammation. In addition, the association of RAGE with HMGB1, TLRs, and TREM-1 in the pathogenesis of insulin resistance in obese diabetic patients has been reported by Subramanian et al. [45]. The AGE-RAGE interaction transduces several signaling pathways involving p21RAS-, p44/p42-MAPK-, PI3K-Akt-, ERK1/2-, JNK-, p38-, PKC-, and NF-κB [23,46,47,48,49]. Finally, the pro-inflammatory molecules IL-1β, IL-6, TNF-α, MCP-1, tissue factor, and VCAM-1 are up-regulated. These oxidative stresses can induce p66shc, a redox sensor, and the enzymes involved in various types of organ and tissue damage under diabetic conditions [50,51]. Biondi et al. [52] have published a comprehensive review on the p66shc redox protein and the emerging complications of diabetes including oxidative stress, cytokines, and cellular aging.

### 3.3. The Cellular Basis of AGE-RAGE Interactions in the Inflammatory Responses of Diabetic Patients

The consequences of hyperglycemia in patients with DM lead to excessive AGE formation and oxidative stress associated with diabetic complications. Gupta et al. [53] found that AGE-PMN interactions up-regulated the NADPH oxidative pathway, resulting in increased ROS generation and oxidative stress. Bansal et al. [54] further investigated the effect of AGE on ROS and reactive nitrogen species (RNS) formation in PMN. The authors also found that AGE could augment PMN-mediated ROS and RNI generation responsible for diabetic complications. Later, Lu et al. [55] demonstrated that MPO and NE released from AGE-activated PMN enhanced differentiation of both Th1 (IFN-γ) and Th17 (IL-17) phenotypes from the naïve CD4+ T cells after AGE-RAGE interactions for immune responses. It is quite interesting that the activated macrophages were the sources of AGEs [56]. Furthermore, dietary AGEs can induce TNF-α production from human macrophages in vitro [57]. AGEs can not only activate human gingival fibroblast and increase production of IL-6 [58] but can also induce cell apoptosis through NLRP3 inflammasome activation [59]. In contrast, our recent investigations revealed that AGE-BSA suppressed Th1/Th2 cytokines but enhanced monocyte IL-6 gene expression via both MAPK-ERK- and MyD88-NF-κΒ p50 signaling pathways [36]. Leerach et al. [60] uniquely demonstrated that RAGE can transport oxytocin from blood into brain for regulating brain functions of social behavior particular the maternal bonding. 

The cellular and molecular basis of AGE-RAGE axis on the immune response and tissue damage are summarized in Figure 3.

## 4. AGE-RAGE Signaling in Mediating the Pathogenesis of Inflamm-Aging

“Inflamm-aging” is defined by Franceschi et al. [61] as the state of chronic sterile low-grade inflammation observed in old organisms. The concept has evolved to a state of broader immune dysregulation in elderly people with persistent increased pro-inflammatory mediator’s production accompanied by a decreased immune response to stimulation [62]. Therefore, the biomarkers of inflamm-aging include chronic activation of innate immunity denoted by increased production of circulatory pro-inflammatory molecules IL-1β, IL-6, TNF-α, and CRP [63,64]. Conversely, adaptive immunity wanes with age, as reflected by reduced responsiveness to vaccination [65,66].

### 4.1. Effects of AGE and RAGE Interaction in the Inflamm-Aging of Various Immune-Related Rheumatic and Autoimmune Diseases

The serum levels of carboxymethyllysine (CML) and methylglyoxal (MGO) are found to be correlated with dietary AGEs intake [43,67], in that 2/3 are deposited in the tissues and 1/3 are excreted by the kidneys [68]. In general, AGEs are regarded as body danger molecules that are capable of binding with RAGE, a pattern recognition receptor [69,70] for mediating inflammatory responses. van Puyvelde et al. [71] have extensively reviewed the effects of AGEs intake on inflamm-aging. Nevertheless, RAGE can bind with a variety of ligands other than AGEs, such as S100/calcium granule protein and high-mobility group protein 1 (HMGB1). The binding can stimulate various cytokine release and acts as a pivotal hub for inducing “inflamm-aging” in different immune-related diseases, including SLE [72,73,74,75], systemic sclerosis [76], autoimmunity in COVID-19 infection [77], and other rheumatic diseases [78]. For unveiling the molecular basis of inflamm-aging induced by AGE, Shen et al. [79] clearly demonstrated that AGE-HSA (human serum albumin) can exert immunosuppressive, inflammatory, and vasculopathic effects in different immune-mediated diseases. Furthermore, the same group noted that the inflammation-related cytokines, including IL-2, IL-6, IL-17, and TNF-α, can accelerate AGE formation in these patients.

### 4.2. The Factors Involving in the Exogenous AGEs Formation and the Interventions for Patients with LSRD

The exogenous sources of AGEs include dietary AGEs, ultraviolet-exposed foods, cigarette smoke compounds, and microwave- and ultrasound-treated foods. In general, the nutrients enriched in proteins and lipids contain higher concentrations of AGEs than in carbohydrates [80]. Accumulation of these molecules would increase damage to tissue/organ structure, consequently triggering the development of age-related diseases. For preventing the damage from exogenous AGEs in the patients with LSRD, Zgutka et al. [81] reported that the lifestyle interventions such as caloric restriction, physical activities, and Mediterranean food with mild cooking may moderate AGE formation, beneficial for a healthy life. 

The definition and the factors contributing to the formation of inflamm-aging are illustrated in Figure 4.

## 5. The Roles of AGE-RAGE Axis in Cancer Risk

It has been demonstrated that the metabolic inclination of cancer cells toward aerobic glycolysis accelerates a hyperglycemic microenvironment for oxidative stress, glycation, and inflammation [82,83]. The formation of AGEs in the microenvironment not only facilitated mutagenesis [84] but also increased protein structure misfolding with loss of functions [85] and finally carcinogenesis [86]. The epidemiological evidence also indicates that T2DM is a risk factor for cancers in general and many site-specific cancers, including colorectal, liver, kidney, uterine, and breast cancers [87].

Gallagher et al. [88] conducted a two-sample Mendelian randomization study to investigate the causal association of T2DM with risk of overall cancer and 22 site-specific cancers. The authors found limited evidence supporting the causal association between fasting glucose level and cancers. However, genetically predicted fasting insulin levels were positively associated with cancers of the uterus, kidney, pancreas, and lung. Pearson-Stuttard et al. [83] further confirmed the potential causal association between the genetically predicted T2DM and fasting insulin concentrations for risk of lung, breast, pancreas, renal, endometrial, and cervical cancers. Obesity, as a sign of pre-diabetes with similar metabolic abnormalities, also exhibits increased risk of cancer and cancer-related mortality [82,83,84].

### 5.1. Molecular Basis of Adipokine and Hyperinsulinemia in Obesity and T2DM Association with Cancer Risk

The abnormal metabolism of adipose tissue in obesity can affect the release of various hormones, adipokines, inflammatory cytokines, growth factors, enzymes, and free fatty acids that may ne implicated in cancer genesis and mortality [85,86]. The cross-talk between adipocytes and cancer cells results in changes of endocrine and paracrine signaling, as reported by Hoy et al. [87]. The hyper-insulinemia can also activate insulin/IGF-signaling with subsequent activation of PI3K/Akt/mTOR and MAPK signaling pathways for promoting cancer growth [88,89,90]. The potential causal effects of T2DM and obesity in cancer risk are shown in Figure 5.

### 5.2. The Roles of AGE-RAGE Axis on Cancer Metabolic and Apoptotic Signaling Pathways for Promoting Cancer Progression

Cancer is a multifactorial disease associated with multiple aberrant signaling pathways different from a non-transformed cell [91]. One of the essential aberrant signals for tumorigenicity is inflammation. Inflammation not only contributes to tumorigenicity but also to tumor promotion by providing bioactive molecules in the tumor microenvironments such as growth factors as well as cell survival factors for preventing cell death, pro-angiogenic factors, and inductive signals for activating epithelial-to-mesenchymal transformation [92,93]. Furthermore, the surrounding infiltrated inflammatory cells release ROS; these are mutagenic, accelerating the genetic instability of the transformed cells toward malignancy [94,95] and sustaining continuous cancer cell proliferation [96]. Kang et al. [97] reported that RAGE is an inducible receptor expressing on the cancer cells. AGE-RAGE binding can activate signaling pathways, including MAPK, ERK1/2, PI3K, AKT, JAK-STAT, and NF-κB for cell survival, inflammation, and cancer progression [98]. These variable aberrant signaling pathways are not only relevant to cancer cell apoptosis, autophagy, and necroptosis, but many other abnormal cell behaviors such as proliferation, invasion/metastasis, angiogenesis, hypoxia aerobic glycolysis, and others. These abnormalities have been comprehensively reviewed by Waghela et al. [99] and Palanissami et al. [100].

### 5.3. The Roles of Dietary Processed Food-Related AGEs on Cancer Risk

A number of studies revealed that the interactions of processed food-related AGEs with RAGE increased oxidative stress, angiogenesis, and inflammatory reaction; these eventually develop into different types of cancers [101]. However, Wada et al. [102] have evaluated the interrelationship between dietary AGE intake and the incidence of total cancer and site-specific cancers in a population-based prospective study in Japan. The daily AGE-CML intake was estimated from a database of CML content in foods determined by ultra-performance LC-tandem mass spectrometry. They failed to observe a significant association between CML intake and total cancer risk in either men or women. However, the potential risk of liver cancer with CML intake needs further investigation.

### 5.4. Implications of RAGE on Predicting the Cancer Incidence in Obesity

The pathobiological mechanisms linking body-mass index and cancer risk had been intensively studied by Renehan et al. [103] and Roberts et al. [104]. It is believed that RAGE belongs to an immunoglobulin superfamily with a type I pattern-recognition receptor property responsible for diabetes [105] and inflammatory diseases [106]. Accordingly, it is quite possible that AGE-RAGE signaling is implicated in the progression from obesity to diabetes and from diabetes to cancer, as reviewed by Garza-Campos et al. [107].

The implications of processed food-related AGEs, RAGE expression, and AGE-RAGE signaling in mediating obesity to diabetes and from diabetes to cancer are demonstrated in Figure 6.

## 6. The Pathological Effects of Other Reducing Sugar-Related Glycation End-Products in Human Diseases

The non-enzymatic binding of reducing sugars to protein molecules via Maillard reaction is initially termed glycated end-products. However, this step of reaction is accelerated especially by glucose-6-phosphate rather than glucose. In fact, the most common sugars eaten by human are not glucose but fructose and sucrose. It has become clear that fructose plays a key role in biochemical alterations and potentially promotes MetS, NAFLD, and T2DM. This is because the liquid form fructose can stimulate the de novo lipogenesis via damage to stimulate hepatic adenosyl-monophosphate dependent kinase activity and increases hepatic triglyceride synthesis, hepatic insulin resistance, and NAFLD, as extensively reviewed by Gugliucci A [108]. Accordingly, it is necessary to investigate the contents of human daily dietary sugars in the sweets and drinks in affecting human health.

### 6.1. The Pathological Roles of Fructose-Related AGEs on Human Disease

Oimomi et al. [109] found that the fluorescence intensity was higher after incubating fructose with BSA when compared with glucose, whereas no significant increase was observed with sorbitol. They concluded that fructose plays an important role in AGE-BSA formation via the polyol pathway. Despite its eight-fold higher reactivity, the contribution of extracellular fructose glycation is much less than glucose because of its low plasma concentration (35 μmol/L) compared to glucose (5 mmol/L). Nevertheless, intracellular fructose concentration is elevated in a number of tissues in diabetic patients because the polyol pathway is quite active, leading to equal concentrations of fructose and glucose for intracellular AGEs formation [110]. Diet contributes significantly to the body fructose levels, especially because high fructose syrup overcomes the bitter tastes of natural fructose [111]. Fructose-mediated AGEs are metabolized in the liver, where it stimulates the de novo lipogenesis. The excessive production of triglycerides contributes to hepatic insulin resistance and dyslipidemia. These adverse effects may play important roles in metabolic and inflammatory diseases, as intensively reviewed by Gugliucci A [112].

### 6.2. The Molecular Basis of Fructose-Related AGE Involving in the Different Degenerative Diseases

Mastrocola et al. [113] discovered that a 60% fructose HFRT diet could evoke CML accumulation in the cell body of pyramid neurons in mice hippocampus. This accumulation activated RAGE-NF-κB signaling pathways and induced widespread reactive gliosis and aberrant mitochondrial respiratory complex activity. In addition, a translocation of glutathione-dependent enzyme glyoxalase-1 (GLO-1) from axon to the cell body of pyramidal neurons has been also observed in HFRT mice after 16 weeks feeding. These abnormalities can induce neurodegenerative diseases after high-fructose intake. The same group also found that fructose-related AGEs up-regulated lipogenesis of gastrocnemius muscle by inhibiting SREBP-1c molecule expression through down-regulation of SREBP-inhibiting enzymes SIRT-1 and increased glycation of SREBP-1 activating proteins SCAP. The decreased expression of myogenic regulatory factors leads to alterations in myofiber composition and reduced mitochondrial activity and muscle strength. Interesting, pyridoxamine can suppress fructose-related AGE formation to counteract these adverse effects [114].

Rai et al. [115] used Sprague–Dawley rats exposed to 20% fructose for 16 weeks to discover that a high level of fructose-AGE occurred in serum and gastrocnemius muscle. These diseased rats displayed increased fasting glycemia, impaired glucose tolerance, decreased skeletal muscle Akt (Ser-473) phosphorylation, and enhanced serum triglyceride levels associated with MGO accumulation and up-regulation of RAGE in gastrocnemius muscle. Inhibition of AGE-RAGE axis by amino-guanidine could effectively decrease AGE accumulation and normalize RAGE expression and dolichyl-diphospho-oligosaccharide-protein glycosyltransferase (DDOST) activity in the gastrocnemius muscle. These results suggest that fructose-induced AGE-RAGE signaling in skeletal muscle attributes to impaired glucose homeostasis. 

Faisal et al. [116] found that human IgG can be glycated by fructose to become fructose-glycated IgG with conformational structure perturbation to form various intermediates and AGEs. These AGE-related molecules dropped off the free lysine and arginine residues. Sotokawauchi et al. [117] found that intracellular formation of fructose-AGE in human umbilical vein endothelial cells stimulated ROS generation and vascular adhesion molecule-1 gene expression. This study suggests that fructose can elicit endothelial cell damage partly via AGE-RAGE axis activation. 

For clinical application, Muraoka et al. [118] reported that fructose- and MGO-induced AGEs altered structural and functional properties of salivary proteins. Therefore, these degenerated proteins can be used for monitoring AGEs in the aging process and evaluation of age-related diseases. 

The pathological effects and the molecular basis of fructose-AGE mediated metabolic changes in neurodegenerative and gastrocnemius muscle are illustrated in Figure 7.

### 6.3. Pathological Effects of D-Ribose- and 2′-Deoxyribose-Induced Protein Glycations on Human Diseases

D-ribose is a natural pentose monosaccharide present in all living cells and their living microenvironments and is involved in several important metabolic pathways. Wei et al. [119] incubated D-ribose with BSA to observe the formation of D-ribose-glycated BSA in comparison with glucose and fructose. They found that the formation rate of D-ribose glycated BSA was much faster than the other reducing sugars. The atomic force microscopic observation revealed that the D-ribose-AGE exists in globular amyloid-like polymers deposits. Furthermore, these amyloid-like aggregates exhibited cytotoxicity to neural cells. The protein ribosylation may occur extracellularly and intracellularly to induce subsequent cognitive impairments in neurodegenerative patients [120]. Furthermore, the same group unveiled that ribose selectively reacted with 17 lysine residues, but no selectivity for lysine residue was found in fructose-glycated BSA [121]. These observations may suggest that ribose exhibits greater and faster glycation ability than fructose for the development of diabetic complications. Recently, Ahmad et al. [122] discovered that the bioavailability rate of D-ribose is much higher than that of glucose in diabetic patients for inducing immunoglobulin G ribosylation.

In addition to glucose, fructose, and ribose, 2′-deoxyribose is one of the most abundant reducing sugars in living organisms for the formation of DNA. The degradation of thymidine by thymidine phosphorylase contributes to 2′-deoxyribose formation. This pentose derivative can react with protein to form glycation intermediates, AGEs, and highly reactive carbonyl species (RCS). Rafi et al. [123] elucidated that 2′-deoxyribose-modified BSA showed an altered secondary structure in α-helix and β-sheet, can cause a band shift in the amide-1 region, and diminished free lysine and arginine content. These findings may imply that 2′-deoxyribose is involved in diabetes and age-related diseases. Alenazi et al. [124] reported that 2′-deoxyribose could perturb IgG structures and lead to AGE formation, causing cellular toxicity and leading to diabetic complications. 

The pathological effects of ribose- and 2′-deoxyribose-glycated AGEs in mediating cell toxicity are depicted in Figure 8.

### 6.4. D-Galacose-Related Protein Glycation in Daily Clinical Practice

Galactose is a C4-epimer of glucose that can combine with glucose to form the disaccharide lactose. The major natural source of dietary galactose is milk and other dairy products. In general, glucose, fructose, and galactose are the essential simple sugars in our diet. D-galactose (D-gal) is a powerful glycation mono-saccharide that can induce oxidative stress, as reported by Bunn et al. [125]. Delwing de Lima et al. [126] demonstrated that galactose can be formed endogenously in human cells [127]. Galactose-related AGEs-induced skin aging has been extensively reviewed by Umbayev et al. [128].

### 6.5. Pathological Effects of Glycosylated-Low Density Lipoprotein (LDL) on Human Diseases

Schleicher et al. [129] found increased levels of glycosylated apoprotein B from LDL molecules in the serum of diabetic patients. This increased lipoprotein glycation can promote covalent binding to vascular structure proteins to enhance oxidative reactions and vascular cell damage [130]. Furthermore, MG-LDL was found to be highly immunogenic, eliciting specific antibodies. Finally, MG-LDL-IgG immune complex deposition in the glomerular basement membrane has been reported by Khan et al. [131]. In addition to being glycated by glucose, LDL also can be ribosylated, as reported by Ahmad et al. [132]. MGO-modified LDL contributed to disease pathogenesis and associated complications, as demonstrated by many authors [133,134]. 

The pathological effects of low-density lipoprotein glycation in diabetic complications are demonstrated in Figure 9.

### 6.6. Increased DNA Glycation in Patients with T2DM

Dutta et al. [135] demonstrated that the DNA glycation reaction was similar to the protein glycation by different reducing sugars, such as glucose, fructose, or ribose. Li et al. [136] further demonstrated structural modifications in the 2′-deoxyguanosine by glyoxal and methylglyoxal via non-enzymatic glycation in an in vitro study. Similarly, Ahmad et al. [137] found that DNA-AGE formed by methylglyoxal and lysine glycation in the presence of Cu^2+^ exhibited genotoxicity and immunogenicity. Ribosylation of calf thymus DNA has been demonstrated by Akhter et al. [138]. The DNA ribosylation acquired immunogenicity for eliciting immune responses [139]. During DNA glycation, the bases in the DNA structure might react with different reducing sugars, including glucose, fructose, and deoxyribose, through the Schiff’s base formation of Armadori products. After further dehydration, cyclization, and isomerization, different AGE products are formed. These DNA-AGEs were characterized by the extent of DNA strand break and base modifications to acquire immunogenicity [101], as found in diabetic patients [140,141]. 

In clinical practice, only two DNA-AGEs are detectable in blood and urine, including N2-(1-carboxyethyl)-2′deoxyguanosine (CEdG) and a cyclic diol from direct addition of MG at 1 position to become N2 of guanine (cMG-dG), which is relative unstable and less suitable as a potential biomarker [142,143]. 

The formation of DNA-AGEs and their pathological effects in diabetic complications are illustrated in Figure 10.

## 7. Unveiling the Molecular Basis of Inflamm-Aging, Vasculopathy and Carcinogenesis Induced by AGE-Albumin

AGEs are not only increased in diabetes, age-related diseases, and chronic renal diseases, but also in various immune-mediated diseases, including SLE, RA, systemic sclerosis, chronic psoriasis, and adult-onset Still’s disease [78,144,145]. It is quite interesting that inflamm-aging is the common feature of these diseases. 

In our previous study, using AGEs from bovine serum albumin (AGE-BSA), Shen et al. [36] found that these degenerated molecules could suppress Th1/Th2 cytokine production while enhancing monocyte pro-inflammatory IL-6 gene expression via MAPK-ERK-mediated and MyD88-mediated-NF-κB signaling pathways. These original observations could become the molecular basis of immunosenescence and inflamm-aging in aged patients. Furthermore, to unveil the molecular basis of inflamm-aging induced by AGE in different immune-related diseases, we used AGE-modified human serum albumin (AGE-HSA) to mimic the clinical situation. Shen et al. [79] clearly demonstrated that AGE-HSA decreased IL-2 production by human Jurkat T cells by suppressing p-STAT3, p-STAT4, and p-STAT6, whereas senescence-associated β-galactosidase (SA-βgal) expression was conversely increased. In contrast, AGE-HSA enhanced CCL-5, IL-8, MIF, and IL-1RA production, but SA-βgal production was conversely suppressed by human THP-1-induced differentiated macrophage-like cells. In addition, the AGE-HSA molecule could abrogate the HSA-induced soluble adhesion molecules (sICAM-1, sE-selectin) and endothelin-1 release from endothelial cells, whereas the SA-βgal expression was conversely enhanced. We further unveiled that the inflammation-related cytokines, including IL-2, IL-6, IL-17, TNF-α, and TGF-β, could accelerate AGE formation in in vitro studies. RAGE and other AGE binding receptors, such as scavenger receptors, seem to be involved in the signals of AGE-HSA induced non-specific immunosuppression, inflamm-aging, and vasculopathy in patients with immune-related diseases. 

The pathological effects and molecular basis of different AGE products, including protein-AGEs, LDL-AGEs, and DNA-AGEs, in mediating inflamm-aging, vasculopathy, and carcinogenesis are summarized in Figure 11.

## 8. Conclusions

Metabolic syndrome, including obesity, hyperlipidemia, and hyperglycemia, can enhance the production of AGEs via non-enzymatic Maillard reaction. Lifestyle-related diseases and immune-related diseases, including SLE, RA, systemic sclerosis, or adult-onset Still’s disease, also increase AGE production. These exogenous or endogenous AGEs can be formed between different reducing sugars (glucose, fructose, galactose, ribose, or deoxyribose) and different molecules (protein, lipoprotein, lipid, or DNA). After binding with their receptors (RAGE or scavenger receptors), the different AGE products can induce different diabetic complications, including CVD, AZD, cancer, and inflamm-aging. It is interesting to note that inflammation-related cytokines can accelerate AGE formation during chronic inflammation. It is worth further investigating the molecular basis of chronic inflammation-induced inflamm-aging in different immune-related diseases. 

## Figures and Tables

**Figure 1 biomedicines-12-01699-f001:**
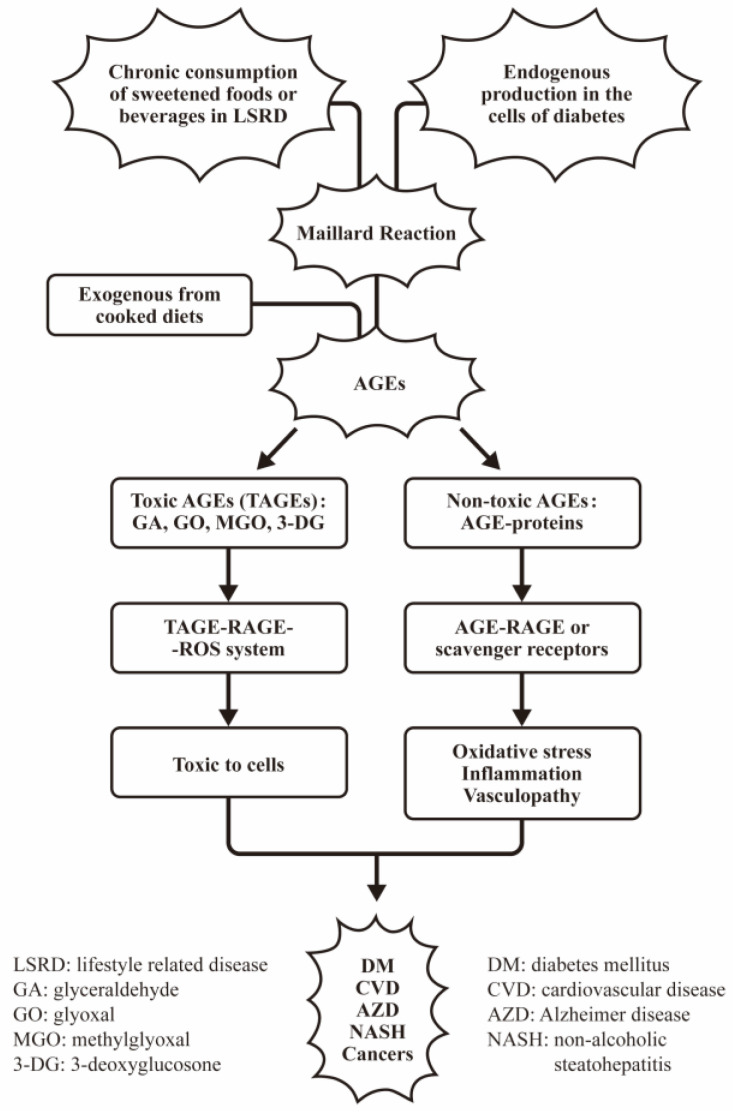
The source, classification, and pathological effects of AGEs on a variety of disease. In general, the AGEs can be divided into toxic AGEs (TAGE) composed of glyceraldehyde (GA), glyoxal (GA), methylglyoxal (MGO), and 3-deoxyglycosone; those are directly toxic to the cells. On the other hand, the non-toxic AGEs are referred to the AGE-proteins exhibiting indirect damage to the cells/tissues via oxidative stress, inflammation, and vasculopathy in different diseases.

**Figure 2 biomedicines-12-01699-f002:**
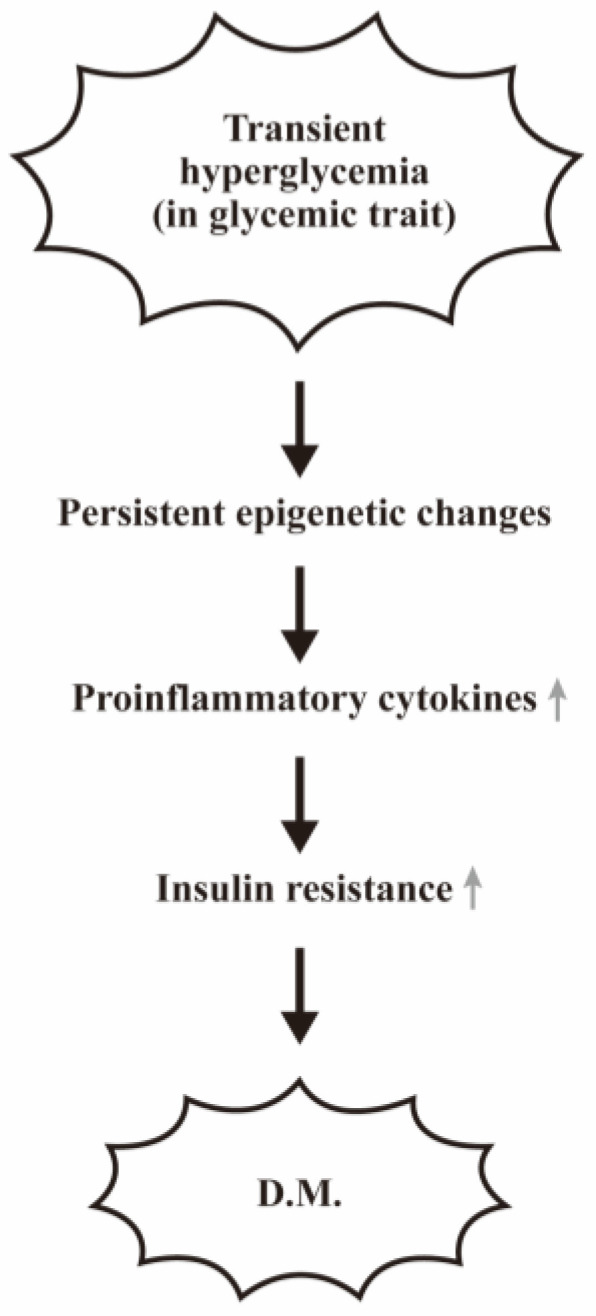
Transient hyperglycemia is a modifiable environmental factor for inducing insulin-resistance in patients with glycemic trait. The molecular basis of transient hyperglycemia in inducing insulin resistance in patients with glycemic trait is postulated that the transient hyperglycemia in glycemic trait can induce persistent epigenetic changes for chronic inflammation and finally insulin resistance. “↑” denotes increase expressions of the molecules or the biological effects.

**Figure 3 biomedicines-12-01699-f003:**
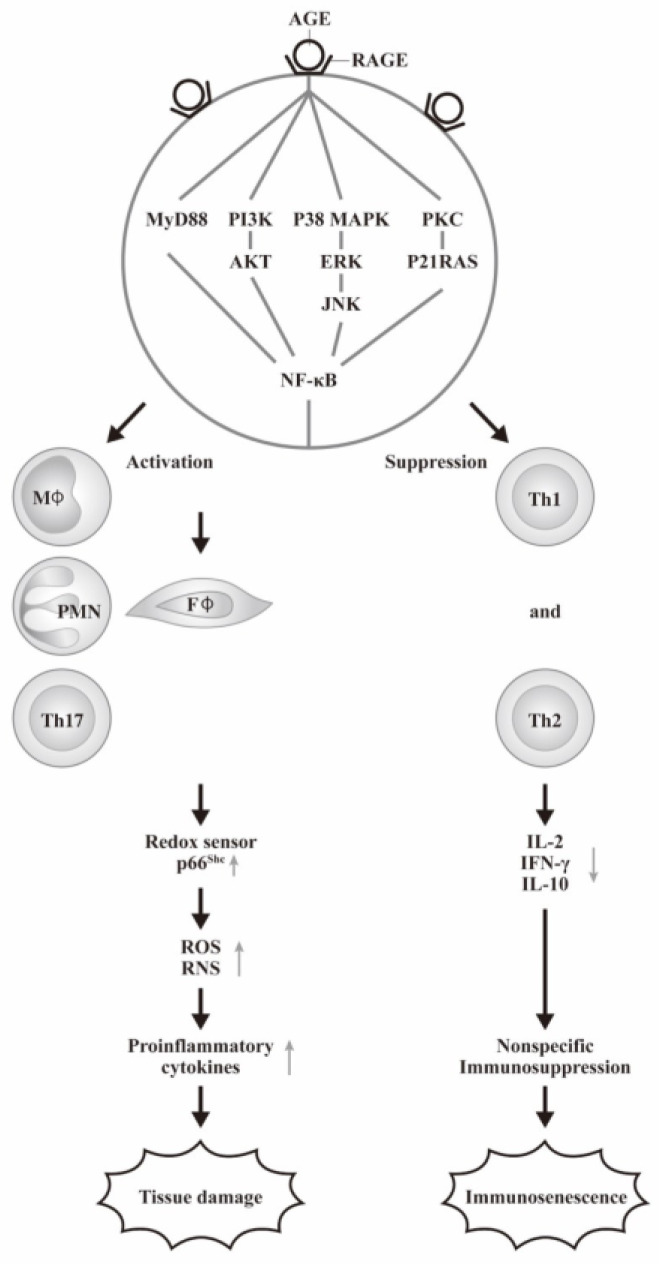
The cellular and molecular bases of AGE-RAGE axis in mediating immune response and tissue damage. AGE-RAGE interaction can transduce MYD88 and PI3K-Akt, p38-MAPK, and PKC-p21-RAS-4 signaling pathways for NF-kB activation. These signals can transduce innate immune cells to increase proinflammatory cytokines production for tissue damage. On the other hand, these signals can concomitantly suppress both Th1 and Th2 cells for immunosenescence. “↑” denotes increase and “↓” denotes decrease expressions of the molecules.

**Figure 4 biomedicines-12-01699-f004:**
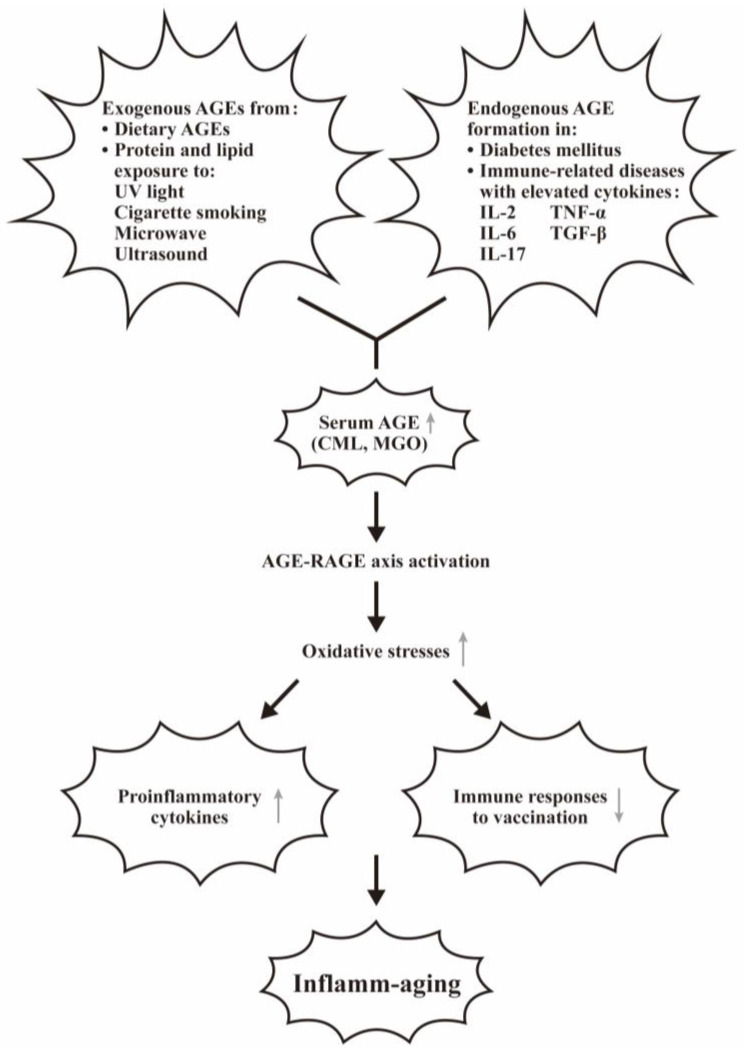
The factors contributing to the exogenous and endogenous AGE formation and induction of inflamm-aging. The exogenous and endogenous AGEs bind with RAGE for enhancing oxidative stresses and proinflammatory cytokine production. These two factors can facilitate inflamm-aging. “↑” denotes increase and “↓” denotes decrease expressions of the molecules or effects.

**Figure 5 biomedicines-12-01699-f005:**
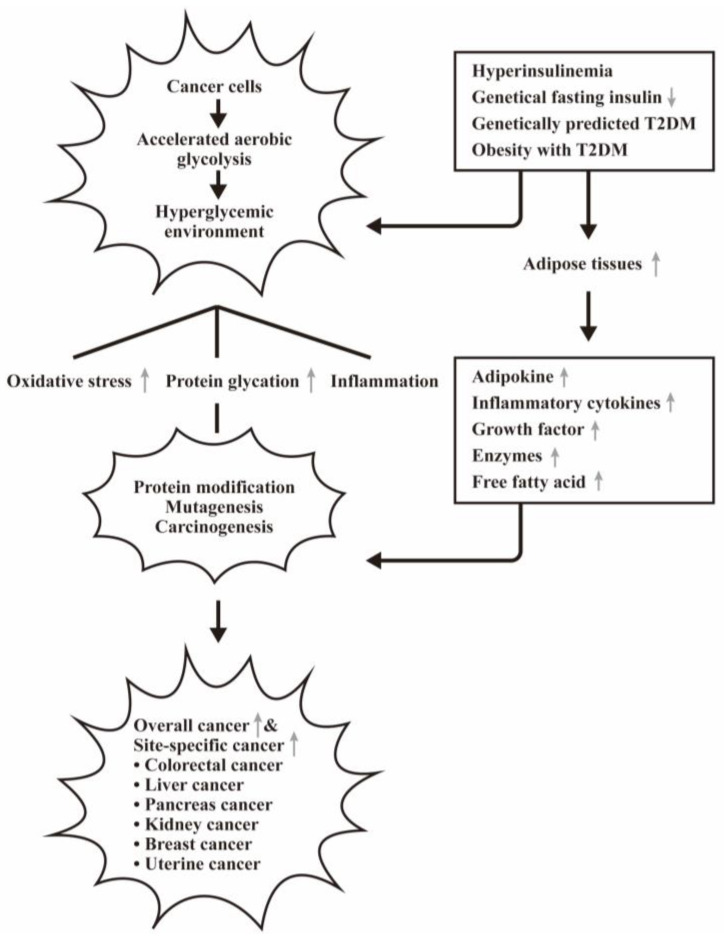
The potential causal effects of type 2 diabetes (T2DM) and obesity in cancer risk. Increased adipose tissues in patients with T2DM and obesity secrete excessive adipokines, inflammatory cytokines, growth factors, and other metabolites. These factors facilitate cancer cells in aerobic glycolysis and the hyperglycemic environment further induces oxidative stress, protein glycation, and inflammation, essential for carcinogenesis. Accordingly, the overall and the site-specific cancers increase in the patients with T2DM and obesity. “↑” denotes increase and “↓” denotes decrease expressions of the molecules or the effects.

**Figure 6 biomedicines-12-01699-f006:**
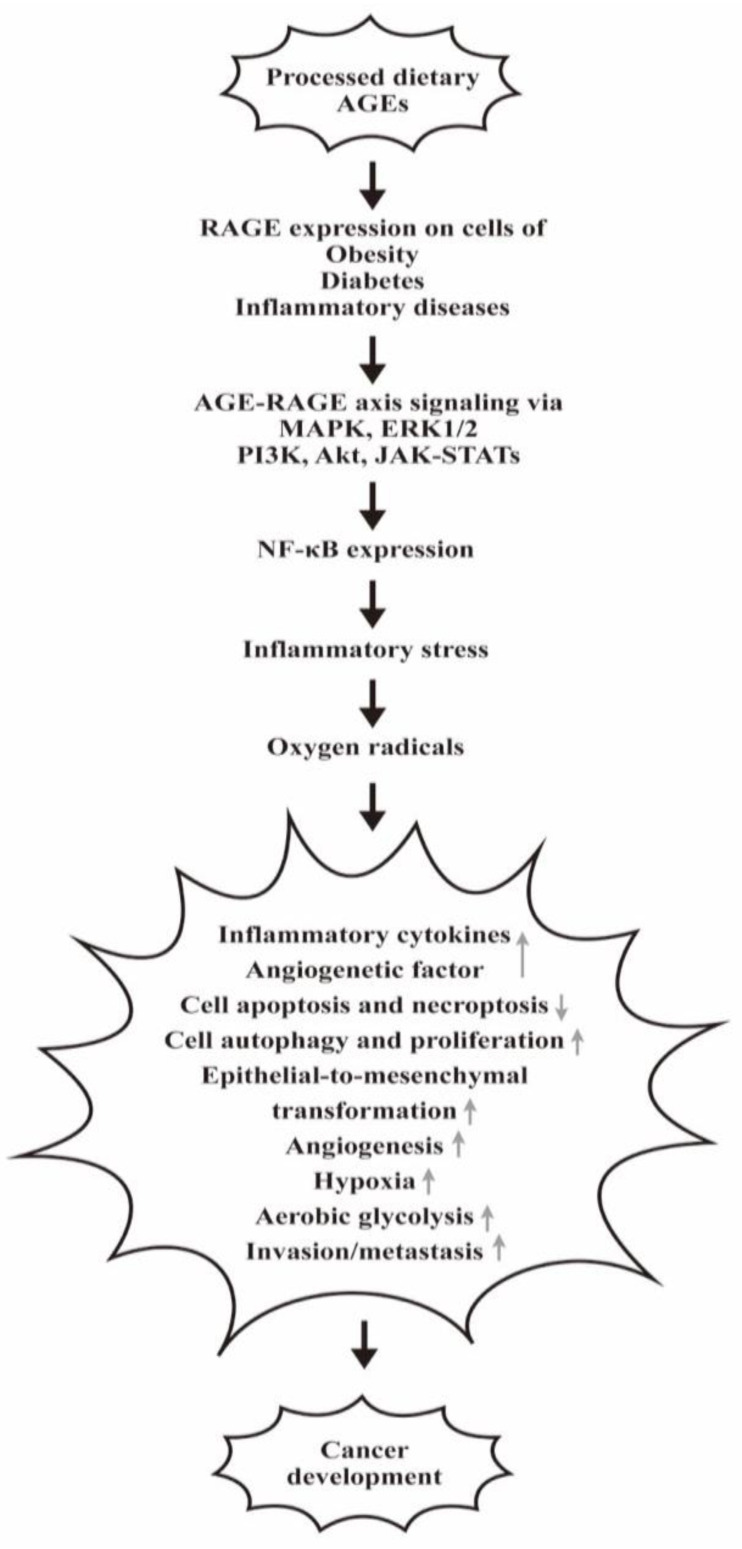
The implication of the processed food-related AGEs in mediating obesity to diabetes and from diabetes to cancer development. The proceeded dietary AGEs via RAGE interact to transduce signaling pathways for inflammation and oxidative stresses in patients with obese and diabetes for carcinogenesis. “↑” denotes increase and “↓” denotes decrease expressions of the molecules or the biological effects.

**Figure 7 biomedicines-12-01699-f007:**
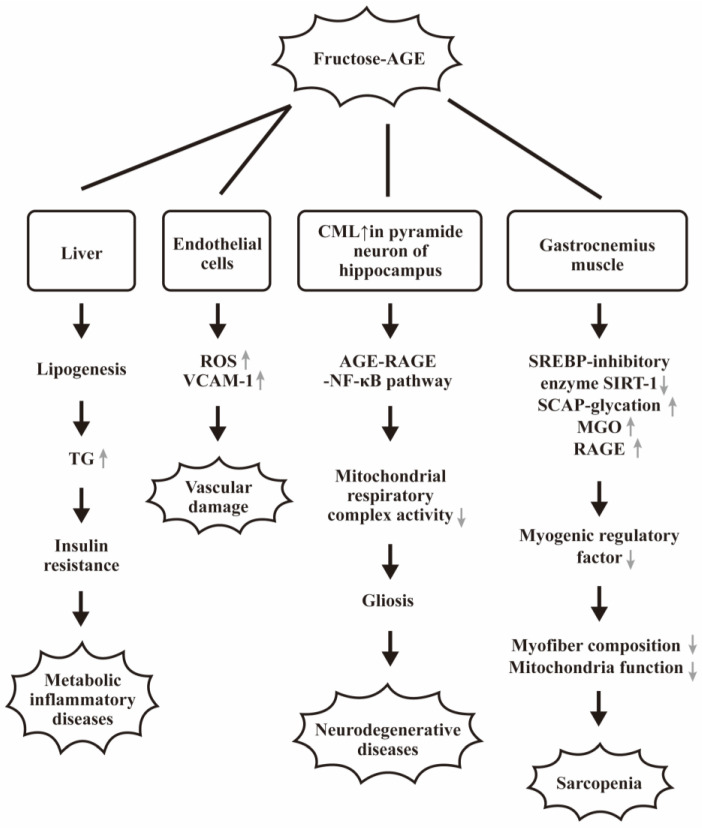
The pathological effects and molecular basis of fructose-AGE mediated metabolic liver inflammation, vascular, and neurodegenerative diseases and sarcopenia. “↑” denotes increase and “↓” denotes decrease expressions of the molecules or the biological effects.

**Figure 8 biomedicines-12-01699-f008:**
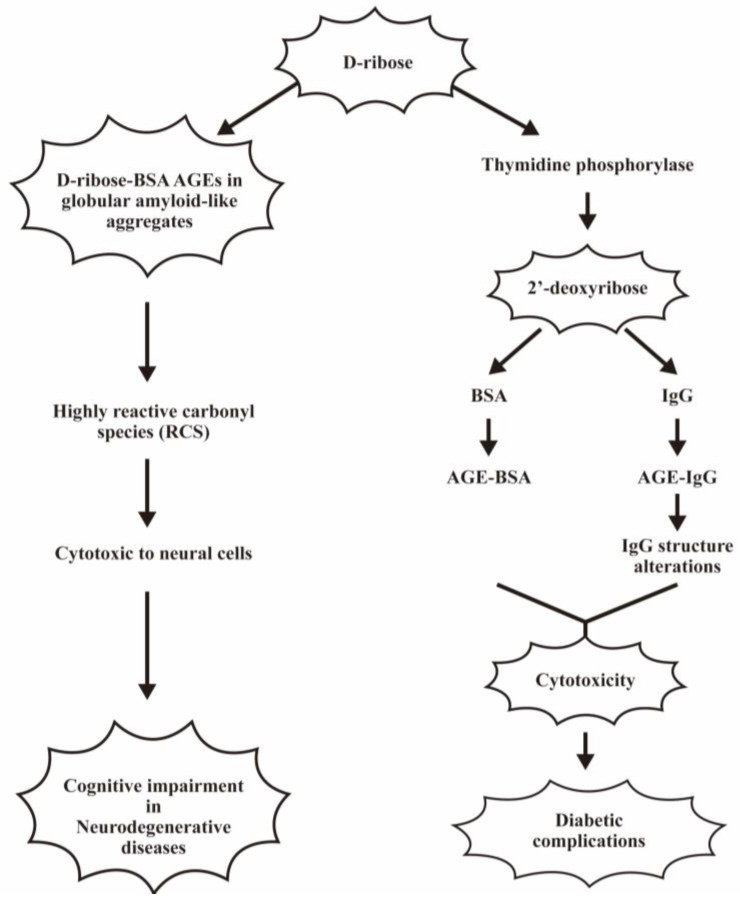
The pathological effects of ribose-related AGEs in the cognitive impairments of patients with neurodegenerative diseases and 2′-deoxyribose related AGEs in diabetic complications. D-ribose-BSA related AGEs can assemble globular amyloid-like aggregates and induce highly reactive carbonyl species (RCS) to damage neural cells. In addition, the 2′-deoxyribose can form AGE-BSA and AGE-IgG to mediate cytotoxicity for diabetic complications.

**Figure 9 biomedicines-12-01699-f009:**
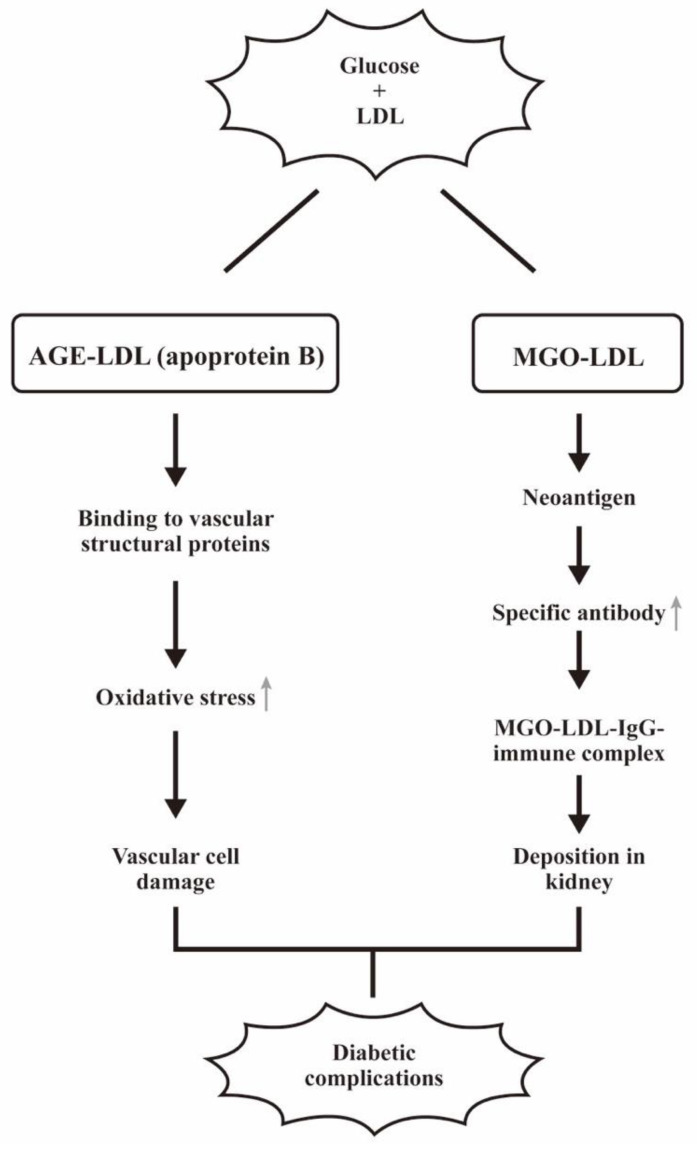
The pathological effects of low-density lipoprotein glycation in diabetic complications. The LDL glycation contains AGE-LDL and MGO-LDL. AGE-LDL with apoprotein can induce vascular cell damage. In contrast, MGO-LDL becomes a neoantigen that can induce specific autoantibodies for MGO-LDL-IgG immune complex deposition in the kidneys. Both pathways can elicit diabetic complications. “↑” denotes increase expressions of the molecules or the biological effects.

**Figure 10 biomedicines-12-01699-f010:**
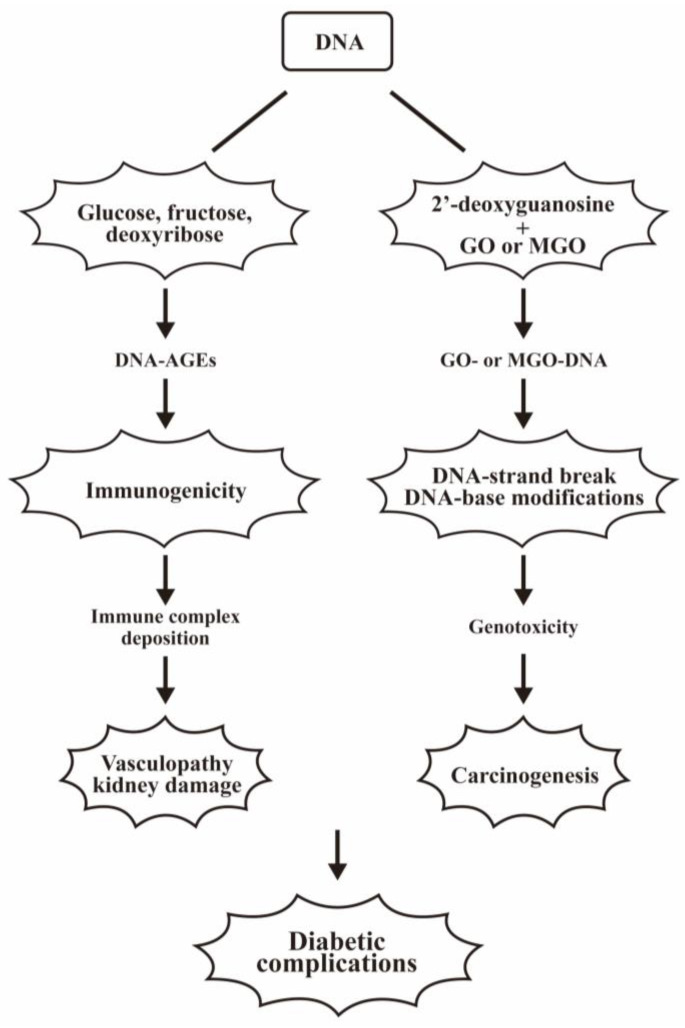
The formation of DNA-AGEs and their pathological effects in diabetic complications. DNAs can be glycated by glucose, fructose, or deoxyribose to become DNA-AGEs. These non-toxic DNA-AGEs are immunogenic and stimulate antibody production and immune complex deposition. The IC deposition inevitably induces vasculopathy and kidney damage. Alternatively, the 2′-deoxyguanosine can bind with GO or MGO to form GO- or MGO-DNAs; these are DNA-strand breakers and DNA-base modifiers. These changes are genotoxic, and induce carcinogenesis in diabetic patients.

**Figure 11 biomedicines-12-01699-f011:**
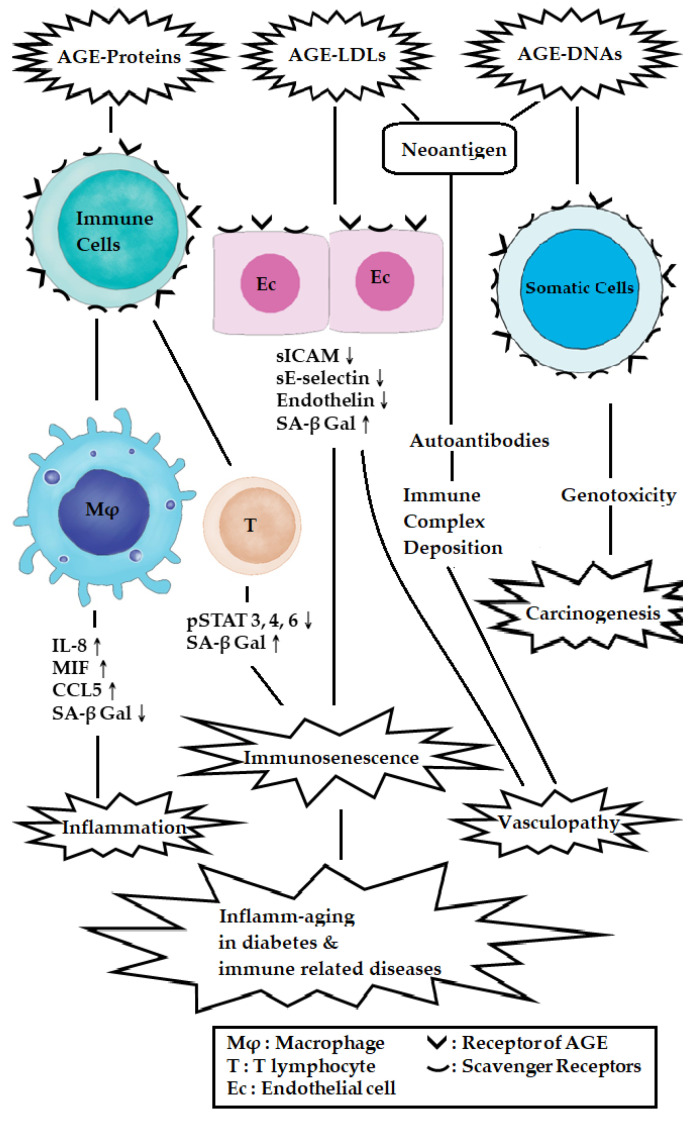
The molecular basis of different AGEs on the pathogenesis of inflamm-aging, vasculopathy, and carcinogenesis in patients with diabetes and immune-related diseases. The protein-AGEs not only activate macrophages to enhance production of pro-inflammatory cytokines (IL-8, MIF, CCL-5) but also increase SA-bgal expression in T cells and endothelial cells, thereby mediating immunosenescence. Furthermore, LDL-AGEs and DNA-AGEs become the neoantigens, eliciting autoantibodies to damage vascular endothelial cells via immune complex doemation. DNA-AGEs have also been found to be genotoxic to somatic cells, thereby inducing carcinogenesis. “↑” denotes increase and “↓” denotes decrease expressions of the molecules.

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
