# Peer review of "Advanced Glycation End-Products Acting as Immunomodulators for Chronic Inflammation, Inflammaging and Carcinogenesis in Patients with Diabetes and Immune-Related Diseases"

_biomedicines, 2024, doi:10.3390/biomedicines12081699_

Round 1

Reviewer 1 Report

Comments and Suggestions for Authors

Authors of the manuscript: biomedicines-3107718 titled "Advanced Glycation End-Products Acting as Immunomodulators for Chronic Inflammation, Inflamm-aging and Carcinogenesis in Patients with Diabetes and Immune-related Diseases" presented a review article that may be useful, but after thorough corrections. First, there is a lack of methods. According to what "key" did the authors select the literature to prepare the manuscript? The review article should be prepared according to: PRISMA.

There is no purpose of research. The summary is poorly prepared and does not reflect the essence of the topic.

Although the authors state in the title that the topic concerns patients with diabetes, in the keywords, there is no word, "diabetes". In addition, the manuscript contains many figures with the mechanisms and role of Advanced Glycation End-Products, but there are no manuscripts that present original research. For example, the authors cite 8 publications (BELOW), all review publications that concern mechanisms and no original research.

18. Vistoli, G.; De Maddis, D.; Cipak, A.; Zarkovic, N.; Carini, M.; Aldini, G. Advanced glycoxidation and lipoxidation end products (AGEs and ALEs): an overview of their mechanisms of formation. Free Radic Res 2013, 47 Suppl 1, 3-27,

doi:10.3109/10715762.2013.815348.

19. Kuzan, A. Toxicity of advanced glycation end products (Review). Biomed Rep 2021, 14, 46, doi:10.3892/br.2021.1422.

20. Twarda-Clapa, A.; Olczak, A.; Bialkowska, A.M.; Koziolkiewicz, M. Advanced Glycation End-Products (AGEs): Formation, Chemistry, Classification, Receptors, and Diseases Related to AGEs. Cells 2022, 11, doi:10.3390/cells11081312.

21. Ott, C.; Jacobs, K.; Haucke, E.; Navarrete Santos, A.; Grune, T.; Simm, A. Role of advanced glycation end products in cellular signaling. Redox Biol 2014, 2, 411-429, doi:10.1016/j.redox.2013.12.016.

22. Sourris, K.C.; Forbes, J.M. Interactions between advanced glycation end-products (AGE) and their receptors in the development and progression of diabetic nephropathy - are these receptors valid therapeutic targets. Curr Drug Targets 2009,

10, 42-50, doi:10.2174/138945009787122905.

23. Aragno, M.; Mastrocola, R. Dietary Sugars and Endogenous Formation of Advanced Glycation Endproducts: Emerging Mechanisms of Disease. Nutrients 2017, 9, doi:10.3390/nu9040385. 544

24. Noriega, D.B.; Zenker, H.E.; Croes, C.A.; Ewaz, A.; Ruinemans-Koerts, J.; Savelkoul, H.F.J.; van Neerven, R.J.J.; Teodorowicz, M. Receptor Mediated Effects of Advanced Glycation End Products (AGEs) on Innate and Adaptative Immunity: Relevance for Food Allergy. Nutrients 2022, 14, doi:ARTN 371 10.3390/nu14020371.

25. Stirban, A.; Gawlowski, T.; Roden, M. Vascular effects of advanced glycation endproducts: Clinical effects and molecular mechanisms. Mol Metab 2014, 3, 94-108,doi:10.1016/j.molmet.2013.11.006.

I suggest removing most of the review works and replacing them with original manuscripts, e.g.

https://pubmed.ncbi.nlm.nih.gov/38965604

https://pubmed.ncbi.nlm.nih.gov/23073299/

and other

It is recommended that the authors prepare a table with the most important original manuscripts on the presented topic.

Reviewer 2 Report

Comments and Suggestions for Authors

This Review reports on the pathogenic role of AGEs in some important lifestyle- and immune-related diseases, focusing on AGEs-indueced biological responses such as, immune-modulation, immunosenescence, inflammation. 

In general, the review is interesting and well orgnized. It is also adequately referenced. The only thing I feel to suggest to authors is to improve Figures. As they are now, they appear basic and poorly prepared.

Reviewer 3 Report

Comments and Suggestions for Authors

The review article discusses the role of Advanced Glycation End-Products Acting as Immunomodulators for Chronic Inflammation, Inflamm-aging, and Carcinogenesis in Patients with Diabetes and Immune-related Diseases. The review is comprehensive, however, there are a few concerns

1. The review is written superficially regarding signaling; like for inflammation, RIS, DM, and CVD, the authors have completely neglected the signaling of RAGE via HMGB-1 and S100 proteins with the involvement of TLRs and TREM-1. The downstream signaling of cytoplasmic kinases has been discussed but ligands and receptors have not been mentioned.

2. The article will benefit by focusing on one or two diseases like inflammation, diabetes, and CVD- the part of metabolic syndrome with CVD. 

3. Inflammation-mediated carcinogenesis is itself a review article and discussing it superficially decreases the interest of the reader. RAGE-mediated carcinogenesis will itself be a separate article. Please consider removing this section.

4. Section 7 is short- please either focus on the pathogenesis or RAGE as a biomarker to diagnose disease. Please consider removing section 7.

5. Please consider changing Figures 1, 3, 5, 6, 7 with schematic diagrams and not the flow charts.

6. Line 50 via RAGE-RAGE-ROS system should be TAGE-RAGE-ROS axis.

In the abstract, it is not clear why the authors wrote this review, please mention the rationale of writing this review.

Round 2

Reviewer 3 Report

Comments and Suggestions for Authors

None